# SHORT REPORT

# Trypanosome bloodstream-specific flagellum attachment proteins can mediate attachment in an insect surface coat environment

Laryssa Vanessa de Liz[1,2,*], Hannah Pyle[2], Patrícia Hermes Stoco[1] and Jack D. Sunter[2,‡]

## ABSTRACT

Throughout the life cycle of the unicellular parasite *Trypanosoma brucei,* its single flagellum remains laterally attached to the cell body by FLA and FLABP proteins, even as the parasite differentiates from the bloodstream form (BSF), found in the mammalian host, to the procyclic form (PCF), in the insect midgut. This differentiation is accompanied by changes in the dominant surface coat protein, from the variable surface glycoprotein to procyclins. There are stage-specific variants of the FLA and FLABP proteins, with FLA2 and FLA2BP found in BSFs, and FLA1 and FLA1BP in PCFs. Yet, how these proteins maintain flagellum attachment during the differentiation from BSFs to PCFs and the accompanying change in surface coat environment is unknown. Here, we used a double-induction system to test whether FLA2 and FLA2BP can maintain flagellum attachment in cells expressing procyclins. Whereas FLA2 compensated for the loss of FLA1, FLA2BP was mislocalised in PCFs and could not compensate for the loss of FLA1BP. Interestingly, when FLA2 was expressed alongside FLA2BP, FLA2BP localised to the flagellum attachment zone and flagellum attachment was maintained. Thus, we conclude that FLA2 and FLA2BP, together, will maintain flagellum attachment as the surface coat environment changes during BSF to PCF differentiation.

KEY WORDS: Trypanosomes, Flagellum, Cell morphogenesis, Differentiation

## INTRODUCTION

*Trypanosoma brucei* is a unicellular parasite with a complex life cycle, alternating between mammalian hosts and tsetse fly vectors. In the mammalian host, bloodstream forms (BSFs) are covered by a dense layer of variant surface glycoproteins (VSGs), essential for evading the immune system (Bangs, 2018; Pays, 2006). When ingested by the tsetse fly, BSFs differentiate in the midgut, into procyclic forms (PCFs). This process involves the replacement of VSGs by GPEET and EP procyclins on their surface (Fenn and Matthews, 2007; Matthews and Gull, 1994; Roditi et al., 1989). As the life cycle progresses, PCFs differentiate into epimastigotes, which are predominantly covered by Brucei alanine-rich proteins

(BARPs) (Casas-Sanchez et al., 2023; Jackson et al., 2013). Finally, epimastigotes differentiate into metacyclic trypomastigotes; once again covered with VSGs (Casas-Sanchez et al., 2023; Ramey-Butler et al., 2015).

The *T. brucei* flagellum is laterally attached to its cell body throughout its entire life cycle (Abeywickrema et al., 2019; Matthews and Gull, 1994; Milne et al., 1998; Wheeler et al., 2013). This attachment is mediated by a specialised cytoskeleton complex known as the flagellum attachment zone (FAZ), which connects the cytoskeleton through the cell body and flagellum membrane to the flagellar cytoskeleton (Sunter and Gull, 2016). There are many proteins localised to the FAZ but only the flagellar adhesion proteins (FLAs) and FLA-binding proteins (FLABPs) have an extracellular domain that is sufficient to span the gap between the flagellum and cell body membranes. Hence, these proteins are essential for flagellum attachment (De Liz et al., 2023; Nozaki, 1996; Sun et al., 2013; Woods et al., 2013).

Notably, whereas most trypanosomatids possess a single FLA and a single FLABP, *T. brucei* has an expansion of these protein families (De Liz et al., 2023). In PCFs, flagellum attachment is maintained by the interaction of FLA1 and FLA1BP, whereas BSFs express FLA2 and FLA2BP (Jensen et al., 2014; Siegel et al., 2011; Sun et al., 2013; Tinti and Ferguson, 2023; Vasquez et al., 2014; Woods et al., 2013). The specific FLA and FLABP expression patterns in other life cycle stages have not been defined, but transcriptomic data indicate different expression levels during tsetse fly infection. FLA1 and FLA1BP are upregulated within the midgut, whereas parasites within the proventriculus express higher levels of FLA2 and FLA1BP. Finally, within the salivary glands, FLA2 and FLA2BP are upregulated (Naguleswaran et al., 2021).

As the parasite is ingested by the tsetse fly, VSGs are released from the cell surface and the parasite starts to express procyclins (Grandgenett et al., 2007; Moreno et al., 2019). Within 2 h, over 90% of cells have procyclins on their surfaces and by 6 h only 1% retain detectable levels of VSGs (Acosta-Serrano et al., 2001; Matthews and Gull, 1994; Roditi et al., 1989). Finally, by 12 h, the differentiation is complete, and the parasites initiate their first cell cycle as PCFs (Gruszynski et al., 2006; Ziegelbauer et al., 1990). Previous studies have shown that the existing FAZ is a stable structure, and during RNA interference (RNAi)-mediated knockdown of FAZ proteins the old flagellum remains laterally attached to the cell body, with only the new flagellum being detached (LaCount et al., 2000; Rotureau et al., 2013; Sun et al., 2013). Moreover, work has shown that FLA1BP is not incorporated into the mature FAZ, aligning with the low or absent turnover of flagellar proteins in the radial spokes, central pair and outer dynein arms in the mature flagellum (Sunter et al., 2015; Vincensini et al., 2018). Therefore, although the surface coat proteins are rapidly replaced during differentiation, it is unlikely that the FLA2–FLA2BP pair will be replaced by FLA1–FLA1BP during the early stages of differentiation. Thus, we proposed a model in which FLA2–FLA2BP continues to maintain flagellum attachment

[1]Departamento de Microbiologia, Imunologia e Parasitologia, Universidade Federal de Santa Catarina, Florianópolis, 88035-972, SC, Brazil. [2]Department of Biological and Medical Sciences, Oxford Brookes University, Oxford, OX3 0BP, UK.
*Present address: University of York, Heslington, York, YO10 5DD, UK.

‡Author for correspondence ( jsunter@brookes.ac.uk)

J.D.S., 0000-0002-2836-9622

Journal of Cell Science

until the completion of differentiation, and after that, as the PCF parasite initiates its first cell division, FLA1–FLA1BP would be incorporated into the new FAZ (De Liz et al., 2023). This would require the FLA2–FLA2BP pair to maintain flagellum attachment in a cell covered by procyclins. Here, using a double-induction system, we demonstrate that FLA2–FLA2BP proteins can maintain flagellum attachment in a procyclin coat environment.

## RESULTS AND DISCUSSION
### FLA2 can compensate for FLA1 loss
To determine whether FLA2 and FLA2BP can maintain flagellum attachment in cells with a procyclin coat, we used a double-induction expression system that allows the independent depletion of one target protein and expression of another (Sunter, 2016). We first generated a cell line capable of inducing FLA1 depletion via RNAi and FLA2 expression (FLA1-RNAi+FLA2 cell line). Cells were grown for 24 h under four conditions: with doxycycline alone to deplete FLA1; with vanillic acid alone to induce FLA2 expression; with both doxycycline and vanillic acid to simultaneously deplete FLA1 and express FLA2; and without any induction. After induction, cells underwent detergent extraction to generate cytoskeletons, which were imaged to assess stable flagellum attachment (Fig. 1A; Fig. S1A).

Since the doubling time of *T. brucei* is ∼8.5 h (Benz et al., 2017), a 24 h induction period allows for 2–3 cell divisions. To assess the ability of FLA2 to maintain flagellum attachment after induction, we focused on G1 cells, characterised by a single nucleus, kinetoplast and flagellum. Three flagellum attachment defects were observed, including complete flagellum detachment, partial

flagellar detachment and loop formation (Fig. S1B; Table S1). For clarity, all three defects were collectively classified as flagellum detachment.

Induction of FLA1 RNAi led to flagellum detachment in most cells, with only 18.7%±5.9 (mean±s.e.m.) having an attached flagellum. FLA1 RNAi does not affect all cells because FLA1–FLA1BP are part of a stable structure; therefore, the disruption of protein synthesis affects only cells undergoing new flagellum and FAZ assembly (Kohl, 2003; Sun et al., 2013). The expression of FLA2 alone (induced with vanillic acid) did not affect flagellum attachment (Fig. 1A,B). Double induction of FLA1 RNAi and FLA2 expression resulted in 99.0%±1.0 of cells maintaining flagellum attachment. FLA1 depletion and FLA2 expression were confirmed by transcriptomic analysis (Fig. S2A). Overall, this demonstrates that FLA2 can compensate for FLA1 loss in PCFs and shows that the sequence differences between FLA1 and FLA2 (∼65% identical) do not disrupt the ability of FLA2 to bind to FLA1BP.

### FLA2BP requires FLA2 to maintain flagellum attachment in procyclic cells
To determine whether FLA2BP can compensate for the loss of FLA1BP in PCFs, we generated a cell line in which we were able to deplete FLA1BP by RNAi and induce the expression of FLA2BP, tagged at the C terminus with a Ty epitope (FLA1BP-RNAi+FLA2BP::Ty cell line). Like FLA1 depletion, FLA1BP knockdown disrupted flagellum attachment in most cells, leaving only 17.3%±1.8 (mean±s.e.m.) with an attached flagellum, whereas the expression of FLA2BP::Ty had no effect (Fig. 2A,B). However, in double-induced cells, only 15.7% ±1.4 of cells had an attached

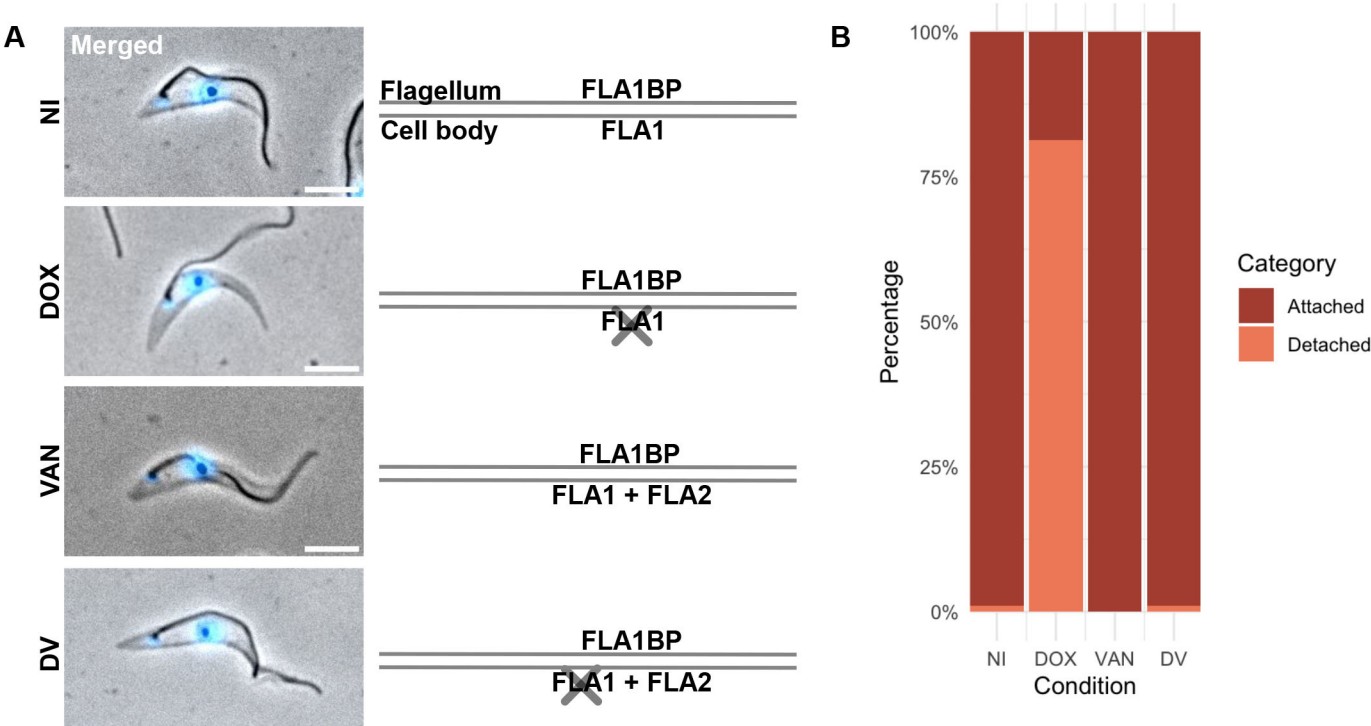

**Fig. 1. FLA2 can compensate for FLA1 loss in PCFs.** (A) On the left, representative images of methanol-fixed detergent-extracted cytoskeletons (FLA1-RNAi+FLA2 cell line) following 24 h of induction. Cells were induced with doxycycline (DOX), vanillic acid (VAN), or both doxycycline and vanillic acid (DV), or were not induced (NI). Hoechst 33342 staining is shown in blue. Scale bars: 5 µm. On the right, the schematic representation shows which FLA and FLABP proteins are expressed in each condition (FLA1BP, FLA1 and FLA2) or depleted (X symbol), with the lines indicating the flagellum and cell body membranes. (B) Percentage of cells as in A with different flagellum attachment and detachment phenotypes from the induction of FLA1BP RNAi and FLA2 expression. The mean of triplicates is shown. N=100 cells per induction condition per replicate.

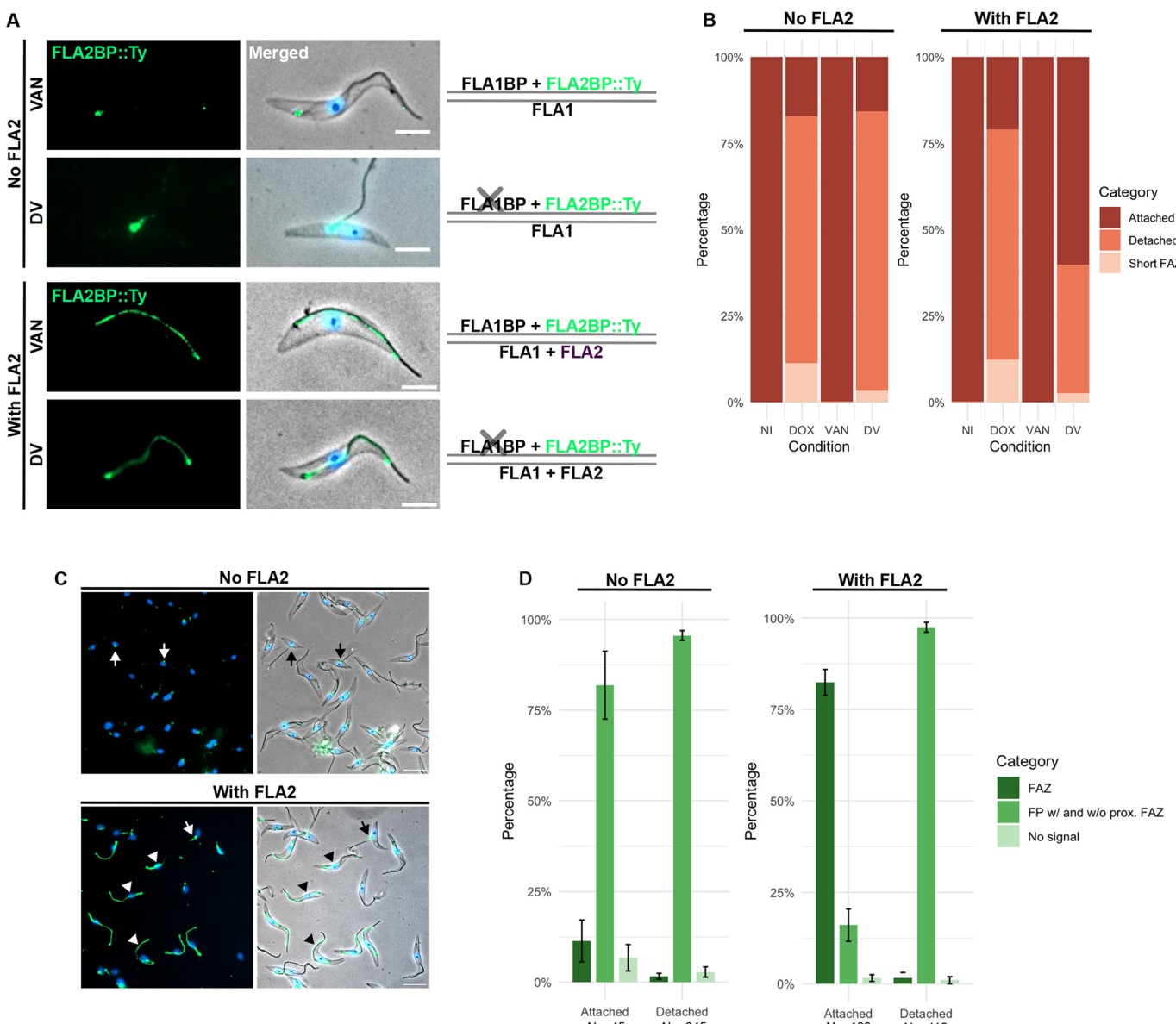

**Fig. 2. Expression of both FLA2BP and FLA2 is required to compensate for FLA1BP loss.** (A) On the left, representative images of immunofluorescence of methanol-fixed detergent-extracted cytoskeletons from either the FLA1BP-RNAi+FLA2BP::Ty cell line (no FLA2) or the FLA1BP-RNAi+FLA2BP::Ty+FLA2 cell line (with FLA2) after 24 h of induction with vanillic acid (VAN) or with doxycycline and vanillic acid (DV). FLA2BP::Ty is shown in green. Hoechst 33342 staining of DNA is shown in blue. Scale bars: 5 µm. On the right, the schematic representation shows which FLA and FLABP proteins are being expressed (FLA1, FLA2, FLA1BP and FLA2BP::Ty) or depleted (X symbol), with the lines indicating the flagellar (top) and cell body (bottom) membranes. (B) Quantification of flagellum attachment and detachment phenotype for cells as in A. Cells were induced with doxycycline (DOX), vanillic acid (VAN), or both doxycycline and vanillic acid (DV), or were not induced (NI). The mean of triplicates is shown. N=100 cells per induction condition per replicate. (C) Fields of view of immunofluorescence of methanol-fixed cytoskeletons of FLA1BP-RNAi+FLA2BP::Ty cell line (no FLA2) or FLA1BP-RNAi+FLA2BP::Ty+FLA2 cell line (with FLA2). Cells were double induced for 24 h (FLA1BP depletion and induced FLA2BP::Ty expression). Overlay of phase-contrast images, FLA2BP::Ty (green) and Hoechst 33342 staining of DNA (blue). Arrows indicate examples of cells with FLA2BP::Ty localised in the flagellar pocket with and without proximal FAZ/flagellum localisation (FP w/ and w/o prox. FAZ in D), while arrowheads show FLA2BP::Ty localised along the FAZ. Scale bars: 10 µm. (D) Distribution of FLA2BP::Ty signal in cells with either an attached flagellum or a detached flagellum for double-induced cells as in C (FLA1BP depletion and induced FLA2BP::Ty expression). The mean of triplicates is shown. The bars indicate the s.e.m. N values represent the accumulated total of cells analysed from three replicates.

flagellum, indicating that FLA2BP was unable to compensate for the loss of FLA1BP. In this experiment, we noted an additional abnormal cell morphology, which was named 'short FAZ'. These cells had a flagellum extending along the cell body but with a shorter FAZ, reduced cell body length and a longer free flagellum (Fig. S1B).

To investigate why FLA2BP failed to compensate for FLA1BP loss, we analysed its localisation in vanillic acid-induced whole

cells by immunofluorescence with the BB2 (anti-Ty) antibody (Fig. S3A). In these cells, FLA2BP was found in the cytoplasm with a bright signal around the flagellar pocket; however, we were unable to discern the FAZ clearly and therefore switched to examining FLA2BP localisation in detergent-extracted cytoskeletons induced with vanillic acid and doxycycline (Fig. 2C,D; Fig. S3B; Table S1). In most cytoskeletons, FLA2BP was restricted to the flagellar

pocket region and proximal flagellum/FAZ, whether the flagellum was detached (95.5% ±0.6; mean±s.e.m.) or attached (82.2% ±1.9), and was only infrequently found along the length of the FAZ. This suggests that FLA2BP was unable to compensate for FLA1BP loss due to its mislocalisation.

We hypothesised that the FLA2–FLA2BP interaction is required for FLA2BP to be correctly positioned along the FAZ. To test this, we modified the FLA1BP-RNAi+FLA2BP::Ty cell line to express FLA2 constitutively (FLA1BP-RNAi+FLA2BP::Ty+FLA2) by replacing the FLA2 3′ untranslated region (3′ UTR) with a truncated version of the aldolase 3′ UTR. Transcriptomic analysis confirmed FLA2 expression (Fig. S2B,D). As expected, inducing FLA1BP RNAi in this cell line led to flagellum detachment in most cells, with only 21.0%±2.6 (mean±s.e.m.) of cells having an attached flagellum. Again, FLA2BP::Ty expression alone did not affect flagellum attachment. Surprisingly, in double-induced cells, flagellum attachment was seen in 60.0% ±3.5 of cells, nearly three times higher when compared to that seen in FLA1BP RNAi-induced cells (Fig. 2A,B). This was associated with a shift in FLA2BP::Ty localisation, with 82.8%±4.8 6 (mean±s.e.m.) of cells with flagellum attachment having FLA2BP::Ty localised along the FAZ, whereas in the 97.3%±2.4 of cells with a detached flagellum, FLA2BP::Ty remained restricted to the flagellar pocket and proximal FAZ/flagellum (Fig. 2C,D).

These findings suggest that FLA2BP needs to bind to FLA2 to localise along the length of the FAZ within the flagellum, and when both proteins are expressed, they can maintain flagellum attachment in procyclin-covered cells when FLA1BP is depleted. There are two types of procyclin (GPEET and EP), and during the early stages of tsetse fly infection, both types are expressed, with GPEETs being predominant (Bütikofer et al., 1997; Knüsel and Roditi, 2013; Naguleswaran et al., 2018; Treumann et al., 1997). Our experiments were conducted in cultured PCFs, which predominantly express GPEET (Fig. S2E). Thus, our results indicate that the FLA2–FLA2BP interaction is compatible with this environment and can support flagellum attachment during differentiation from BSFs to PCFs.

Previous studies investigating the FLA1–FLA1BP interaction in PCFs have shown that in cells with flagellum detachment due to depletion of FLA1, FLA1BP localises to the FAZ within the flagellum (Sun et al., 2013). Likewise, the lack of the FLA1BP extracellular domain, responsible for its interaction with FLA1, does not affect its localisation to the FAZ within the flagellum. However, in both the above cases, the protein signal is predominantly in the proximal region of the flagellum, like the signal we saw with FLA2BP::Ty when FLA1BP was depleted and FLA2 was not expressed (Rotureau et al., 2013; Sun et al., 2013). Thus, further experiments will be needed to determine whether FLA1BP also requires FLA1 binding for localisation along the length of the FAZ in the flagellum.

An intriguing question is why *T. brucei* evolved multiple FLA–FLABP pairings. Previously, we have shown that the duplication of FLA and FLABP is only seen in *T. brucei* and *T. congolense*, and we have postulated that this enables these parasites to maintain three distinct major surface coat proteins (De Liz et al., 2023). Here, we show that FLA2 and FLA2BP can operate in a cell covered by procyclins for an extended period, suggesting that these FLA–FLABP pairings do not require a specific coat environment in which to work, though it is currently unclear whether FLA1–FLA1BP can operate in a cell covered by VSGs. However, these experiments were done *in vitro*, so the consequence of expressing the FLA2–FLA2BP pairing while in the tsetse fly is unknown. Together, this points to a potential tuning phenomenon, with each pairing tuned to operate most effectively within that coat environment and in the context of the specific extracellular environment in the tsetse fly or mammalian host.

## The intracellular domain of FLA1BP is insufficient for FLA2BP FAZ localisation

Given that the intracellular domain of FLA1BP has been shown to be important for its localisation to the FAZ within the flagellum in *T. brucei* PCFs (Sun et al., 2013), and the FLA2BP intracellular domain only shares 59% identity with that of FLA1BP, we investigated whether replacing the FLA2BP intracellular domain with the FLA1BP intracellular region could enable correct localisation and function of the FLA2BP extracellular region. To test this, we generated a cell line allowing the depletion of FLA1BP by RNAi and the expression of FLA2BP extracellular region fused to the transmembrane and intracellular domains of FLA1BP, named FLA2BP$^E$::FLA1BP$^I$::Ty (FLA1BP-RNAi+FLA2BP$^E$::FLA1BP$^I$:: Ty cell line; Fig. 3A).

In this cell line, FLA1BP depletion resulted in only 17.7%±0.7 (mean±s.e.m.) of G1 cells with an attached flagellum, whereas the expression of FLA2BP$^E$::FLA1BP$^I$::Ty had little impact on flagellum attachment (Fig. 3B,C). More importantly, most double-induced cells had a detached flagellum, with only 14.3%±2.6 of cells having an attached flagellum. This suggests that the intracellular region of FLA1BP is insufficient for the FLA2BP extracellular domain to function in PCFs. As observed with FLA2BP::Ty, this lack of functionality is likely due to the incorrect localisation of this mutant, since in 96.0%±2.6 (mean±s.e.m.) of double-induced cells with a detached flagellum, FLA2BP$^E$::FLA1BP$^I$::Ty was confined to the flagellar pocket region and proximal FAZ/flagellum (Fig. 3D,E).

Given that FLA2 expression influenced FLA2BP localisation and function, we modified the FLA2 3′ UTR region of the FLA1BP-RNAi+FLA2BP$^E$::FLA1BP$^I$::Ty cell line to enable its constitutive expression, generating the FLA1BP-RNAi+FLA2BP$^E$:: FLA1BP$^I$::Ty+FLA2 cell line. FLA2 expression was confirmed via transcriptomics (Fig. S2C,D). Upon FLA1BP RNAi induction, flagellum detachment was observed in most cells, with only 23.7%±2.9 (mean±s.e.m.) having an attached flagellum (Fig. 3B,C). As expected, the induction of FLA2BP$^E$::FLA1BP$^I$:: Ty alone did not affect flagellum attachment. Notably, in double-induced cells there was an increase in flagellum attachment compared to FLA1BP RNAi induction alone (54.0%±2.0 of cells retained attachment). This indicates that FLA2 expression facilitated flagellum attachment in this mutant. Consistently, 70.4%±3.2 (mean±s.e.m.) of cells with an attached flagellum had FLA2BP$^E$::FLA1BP$^I$::Ty localised along the FAZ, whereas in 70.1%±6.3 of the cells with a detached flagellum, FLA2BP$^E$::FLA1BP$^I$::Ty was restricted to the flagellar pocket region and proximal FAZ/flagellum (Fig. 3D,E).

Overall, our results show that FLA2 and FLA2BP can maintain flagellum attachment in a cell covered by procyclins and will therefore maintain flagellum attachment during BSF to PCF differentiation. Moreover, our findings show that the flagellar pocket is a key site for FAZ assembly. The different membrane domains within the flagellar pocket region of the cell are defined by the collarette (flagellum–flagellar pocket) and the flagellar pocket collar (flagellar pocket–cell body) (Gadelha et al., 2009) (Fig. 4). The flagellar pocket collar also acts to bring the flagellum and cell body membrane into close proximity, facilitating FAZ assembly. Our data support this, as FLA2BP and FLA2BP$^E$::FLA1BP$^I$ were restricted to the flagellar pocket region without FLA2 expression and were only localised along the flagellum FAZ domain when FLA2 was expressed. We propose that the interaction of a FLABP extracellular domain with the extracellular domain of a FLA likely

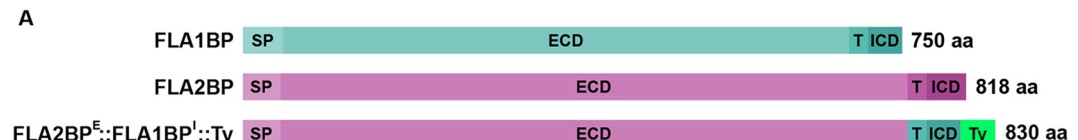

**Fig. 3. The intracellular domain of FLA1BP does not affect FLA2BP localisation and function.** (A) Schematic representation of FLA1BP, FLA2BP and FLA2BP$^E$::FLA1BP$^I$::Ty domains (aa, amino acids; ECD, extracellular domain; ICD, intracellular domain; SP, signal peptide; T, transmembrane domain; Ty, double Ty tag). (B) On the left, representative images of immunofluorescence of methanol-fixed detergent-extracted cytoskeletons from either the FLA1BP-RNAi+FLA2BP$^E$::FLA1BP$^I$::Ty cell line (no FLA2) or FLA1BP-RNAi+FLA2BP$^E$::FLA1BP$^I$::Ty+FLA2 cell line (with FLA2) after 24 h of induction with vanillic acid (VAN) or with doxycycline and vanillic acid (DV). FLA2BP$^E$::FLA1BP$^I$::Ty is shown in green. Hoechst 33342 staining of DNA is shown in blue. Scale bars: 5 μm. On the right, the schematic representation shows which FLA and FLABP proteins are being expressed (FLA1, FLA2, FLA1BP and FLA2BP$^E$::FLA1BP$^I$::Ty) or depleted (X symbol), with the lines indicating the flagellar (top) and cell body (bottom) membranes. (C) Quantification of flagellum attachment and detachment phenotype for cells as in B. Cells were induced with doxycycline (DOX), vanillic acid (VAN), or both doxycycline and vanillic acid (DV), or were not induced (NI). The mean of triplicates is shown. N=100 cells per induction condition per replicate. (D) Fields of view of immunofluorescence of methanol-fixed cytoskeletons from either the FLA1BP-RNAi+FLA2BP$^E$::FLA1BP$^I$::Ty cell line (no FLA2) or the FLA1BP-RNAi+FLA2BP$^E$::FLA1BP$^I$::Ty+FLA2 cell line (with FLA2). Cells were double induced for 24 h (FLA1BP depletion and induced FLA2BP$^E$::FLA1BP$^I$::Ty expression). Overlay of phase-contrast images, FLA2BP$^E$::FLA1BP$^I$::Ty (green) and Hoechst 33342 staining of DNA (blue). Arrows indicate examples of cells with FLA2BP$^E$::FLA1BP$^I$::Ty localised in the flagellar pocket with and without proximal FAZ/flagellum localisation (FP w/ and w/o prox. FAZ in E), while arrowheads show FLA2BP$^E$::FLA1BP$^I$::Ty localised in the FAZ. Scale bars: 10 μm. (E) Distribution of FLA2BP$^E$::FLA1BP$^I$::Ty signal in cells with either an attached flagellum or a detached flagellum for double-induced cells as in D (FLA1BP depletion and induced FLA2BP$^E$::FLA1BP$^I$::Ty expression). The mean of triplicates is shown. The bars indicate the s.e.m. N values represent the accumulated total of cells analysed from three replicates.

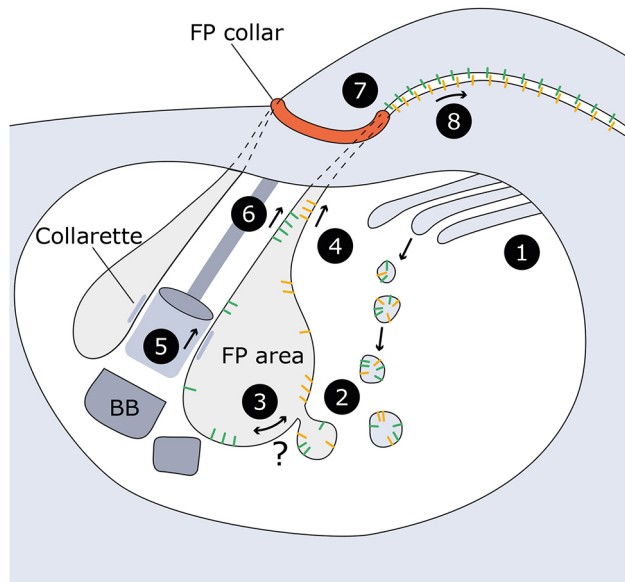

**Fig. 4. Model of FLA and FLABP assembly into the FAZ.** FLAs (yellow) and FLABPs (green) traffic through the Golgi (1). Vesicle trafficking (2) ensures delivery of FLAs and FLABPS to the flagellar pocket (FP), where an unknown sorting mechanism guides FLAs and FLABPs towards different pathways (3), although it is possible this sorting mechanism occurs before the flagellar pocket, with distinct docking sites for the different proteins. FLAs move towards the cell body membrane (4), while FLABPs cross the transition zone and collarette (5) and move along the flagellum membrane (6). Around the flagellar pocket collar (FP collar; red), FLAs and FLABPs interact (7), enabling integration of the FLA–FLABP pair into the elongating new FAZ (8). BB, basal body. Based on Absalon et al., 2008.

occurs as the flagellum and cell body membranes are brought together by the flagellar pocket collar, which enables the effective integration of this FLA–FLABP pairing into the FAZ (Fig. 4). This highlights the complexity of coordinating new FAZ assembly in the proximal region of the growing new flagellum and points to the flagellar pocket as key staging post for the integration of new FAZ components.

## MATERIALS AND METHODS
### Plasmid constructs
For the RNAi, plasmids containing 716 bp (nucleotides 646–1361) or 299 bp (nucleotides 1–311) fragments from *T. brucei* *Tb*FLA1BP or *Tb*FLA1 genes were cloned into pQuadra, previously described (Inoue et al., 2005).

For the induced expression of the other constructs, the p1271 plasmid was utilised (Sunter, 2016). The recode sequence of FLA1BP was obtained from Twist Bioscience and used to generate the FLA1BP-recode_1271 plasmid. The sequences from FLA2BP and FLA2 were amplified from *T. brucei* strain 427 gDNA and were 98.4% and 99.7% identical, respectively, to FLA2BP-2 (Tb927.5.4580) and FLA2-1 (Tb927.8.4060) from the *T. brucei* TREU927 strain. These were PCR amplified with primers containing *Hind*III and *Xba*I (FLA2BP) or *Hind*III and *Bgl*II (FLA2) sites with ROCHE Expand High Fidelity PCR System, purified with Monarch DNA gel extraction kit and digested with the respective enzymes (NEB). Following purification with Monarch PCR & DNA Cleanup Kit, the digested product was cloned into pJ1271 plasmid previously digested with *Hind*III and *Xba*I or *Hind*III and *Bgl*II, originating FLA2BP::Ty_1271 (double-tagged with Ty) and FLA2_1271.

The construct FLA2BP^E::FLA1BP^I::Ty was generated by fusion PCR, as described previously (Dean et al., 2015). The sequence corresponding to the extracellular region of FLA2BP was amplified from FLA2BP::Ty_1271, while the transmembrane and the intracellular region of FLA1BP were amplified from FLA1BP-recode_1271. The purified PCR products were

used in the following PCR, and the product was purified, digested with *Hind*III and *Xba*I, and cloned into p1271.

The cloning of the constructs was confirmed by digestion with specific restriction enzymes. Additionally, the FLA2BP^E::FLA1BP^I::Ty construct was confirmed by sequencing with the Mix2Seq Kit from Eurofins.

### Cell culture
#### Cell maintenance and generation of cell lines
Parasites were cultivated at 28°C in SDM-79 medium with 10% foetal calf serum, with three weekly subcultures (Brun and Schönenberger, 1979). The cell line *T. brucei* TREU927, previously transfected with the pJ1173 plasmid (Sunter, 2016), was used to generate all cell lines and was authenticated during this study by mRNA sequencing. Cell lines were monitored for contamination, including mycoplasma contamination, through DNA staining and microscopy. The parental cell line was transfected with FLA1_pQuadra or FLA1BP_pQuadra. Once recovered, these cell lines underwent another round of transfection with plasmids FLA2_1271, FLA2BP::Ty_1271, or FLA2BP^E::FLA1BP^I::Ty_1271. For transfections, plasmids were linearised with *Not*I, precipitated with sodium acetate and transfected as described previously (Dean et al., 2015). Cells were recovered in SDM-79 for 8 h before selection with phleomycin (5 μg/ml), blasticidin (20 μg/ml) or hygromycin (25 μg/ml).

The cell lines FLA1BP-RNAi+FLA2BP::Ty and FLA1BP-RNAi+FLA2BP^E::FLA1BP^I::Ty were further modified to allow the constitutive expression of FLA2 as described previously (Dean et al., 2015). For this, the intergenic sequence between the gene encoding the fluorescent protein, the G418 coding sequence and its 3′ UTR was amplified from the pPOT-G418 plasmid using primers that contain sequences homologous to 3′ end of the FLA2 gene and the 5′ end of the FLA2 3′ UTR, enabling displacement of the endogenous FLA2 3′ UTR. The product was precipitated by sodium acetate precipitation and transfected into different cell lines.

### Double-induced protein expression
For each cell line, $1 \times 10^6$ parasites/ml were transferred into flasks. Each flask was incubated with distinct treatments: only doxycycline (1 μg/ml); only vanillic acid (250 μM); both doxycycline and vanillic acid; and one flask without any treatment, referred to as non-induced. Following the 24 h of induction, parasites were harvested for further analysis. Each induction experiment was performed in triplicate.

### Immunofluorescence
Cells and cytoskeletons were obtained following established procedures (Dean and Sunter, 2020). Parasites were harvested by centrifugation at 800 $g$ for 3 min, washed twice with Voorheis modified phosphate-buffered saline (vPBS), and resuspended in 200 μl of vPBS. Then, 40 μl of parasite suspension was allowed to adhere to the slide for 5 min and the excess solution was removed. For cells, parasites were fixed with 4% paraformaldehyde in PBS for 15 min, before quenching with 1% glycine in PBS for 5 min and permeabilisation with PEME [100 mM PIPES-NaOH (pH 6.9), 2 mM MgSO_4, 2 mM EGTA (pH 8), 0.1 mM EDTA (pH 8)] containing 0.1% IGEPAL CA-630 for 5 min. For cytoskeletons, the parasites were incubated for 30 s with a PEME–1% IGEPAL CA-630 solution, followed by 20 min of incubation in methanol at −20°C.

After fixation, slides were transferred to PBS at room temperature for 20 min, blocked with 1% bovine serum albumin (BSA) for 1 h and then incubated overnight at 4°C with anti-Ty antibody BB2 (1:10 in PBS containing 1% BSA; Bastin et al., 1996). Following incubation, the slides were washed with PBS and incubated with goat anti-mouse IgG Alexa Fluor 488 (1:200 in 1% BSA; Invitrogen, A-11001; RRID: AB_2534069) for 1 h at room temperature. Finally, slides were washed, incubated with 0.1 μg/ml Hoechst 33342 for 10 min, washed with PBS and mounted with Vectashield antifade mounting medium. Imaging was conducted using a Zeiss imager Z2 fluorescence microscope with an ORCA Flash4 camera and 63× oil immersion objective. Image acquisition and analysis were performed using Zen Blue software (Zeiss) and ImageJ (NIH, Bethesda, MD, USA), respectively.

## Transcriptomics

A 10 ml volume of $1\times10^6$ parasites/ml were double induced with doxycycline and vanillic acid for 24 h. Non-induced parasites were used as controls. Following 24 h, parasites were harvested by centrifugation at 800 *g* for 7 min and washed with DMEM without serum (ThermoFisher). RNA was extracted using the RNeasy kit (Qiagen) and stored at −80°C. For RNA sequencing (RNA-seq), mRNA was enriched using polyA selection and then sequenced using 100 bp paired-end sequencing (BGISEQ). To quantify transcript abundance, fastq reads were mapped to the *T. brucei* TREU927 transcriptome using Burrows–Wheeler aligner-MEM (v.0.7.17) with default settings (Li and Durbin, 2010). Reads per kilobase per million (RPKM) were calculated using the idxstats in SAMtools (Li et al., 2009).

## Study design

Sample size is reported in the figure legends. All experiments were performed three times unless specified in the figure legends. No data were excluded from the analyses. No blinding or randomisation occurred during these studies.

### Acknowledgements

We thank the Oxford Brookes University Centre for Bioimaging for their support.

### Competing interests

The authors declare no competing or financial interests.

### Author contributions

Conceptualization: J.D.S.; Formal analysis: L.V.d.L., H.P., J.D.S.; Funding acquisition: L.V.d.L., P.H.S., J.D.S.; Investigation: L.V.d.L., H.P.; Project administration: P.H.S., J.D.S.; Resources: J.D.S.; Supervision: P.H.S., J.D.S.; Validation: L.V.d.L.; Writing – original draft: L.V.d.L., P.H.S., J.D.S.

### Funding

L.V.d.L. was a recipient of a Coordenação de Aperfeiçoamento de Pessoal de Nível Superior (CAPES) scholarship, a Brazilian Government Agency. L.V.d.L. received a travel grant from The Company of Biologists, sponsored by Journal of Cell Science. Work in the laboratory of J.D.S. is supported by the Wellcome Trust (221944/Z/20/Z), Leverhulme Trust and Oxford Brookes University. Open Access funding provided by Oxford Brookes University. Deposited in PMC for immediate release.

### Data and resource availability

RNA-seq data are available via the European Nucleotide Archive under BioProject accession PRJEB93833. Resources are available from the authors by request. All other relevant data and details of resources can be found within the article and its supplementary information.

### First Person

This article has an associated First Person interview with the first author of the paper.

### Peer review history

The peer review history is available online at https://journals.biologists.com/jcs/lookup/doi/10.1242/jcs.264370.reviewer-comments.pdf

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
