## [Peer Review File · Journal of Cell Science]

Trypanosome bloodstream-specific flagellum attachment proteins can mediate attachment in an insect surface coat environment

Laryssa Vanessa de Liz, Hannah Pyle, Patr cia Hermes Stoco and Jack D. Sunter

DOI: 10.1242/jcs.264370

Editor: Lotte Pedersen

Review timeline

Original submission:	8 August 2025
Editorial decision:	1 September 2025
First revision received:	2 December 2025
Accepted:	3 December 2025

Original submission

First decision letter

MS ID#: jcs.264370

MS TITLE: Trypanosome bloodstream-specific flagellum attachment proteins can operate in an insect surface coat environment

AUTHORS: Laryssa Vanessa de Liz; Patr cia Hermes Stoco; Jack D Sunter

ARTICLE TYPE: Short Report

Dear Dr Sunter,

We have now reached a decision on the above manuscript.

To see the reviewers' reports and a copy of this decision letter, please go to: View Reviewer Comments

As you will see, the reviewers gave favourable reports but raised some critical points that will require amendments to your manuscript. I hope that you will be able to carry these out because I would like to be able to accept your paper, depending on further comments from reviewers.

Reviewer 1

Advance summary and potential significance to field

This is an excellent paper, which focuses on a relatively unexplored aspect of the biology of flagellum attachment in trypanosomes: why does *Tbrucei*, unusually amongst its relatives, encode two pairs of FLA-FLABP complexes during the life cycle? The authors perform a clear and comprehensive series of experiments that demonstrate rigorously that FLA2-FLA2BP, which normally operates in flagellum attachment in bloodstream form (mammalian stage) *T. brucei* can also operate in procyclic (tsetse stage) *T. brucei*. Furthermore, they provide evidence that the two FLA-FLABP complexes interact separately: ie there is specific and non-interchangeable interactions between FLA1-FLA1BP and FLA2-FLA2BP.

Comments for the author

I have no issues with any of the experiments presented, or with the clearly articulated conclusions. Instead, I congratulate the authors on a very nice study that is, in my view, suitable for publication. However, if they authors had the space and time, perhaps they could add a line or two about a question that is not addressed in this nice work: if FLA2-FLA2BP can functionally replace FLA1-FLA1BP in flagellum attachment in PCF cells, why has *T. brucei* evolved two forms of this complex?

Reviewer 2*Advance summary and potential significance to field*

This manuscript presents a study of proteins of the flagellum attachment zone, the structure critical for attachment of the flagellum to the cell body in the parasitic flagellate *Trypanosoma brucei*. The study focuses on pairs of interacting membrane proteins and addresses whether the pair specifically expressed in the bloodstream form of the parasite can operate in the membrane environment of a tsetse fly form, which is expected to happen at early stages of fly infection. The experiments provide clear evidence that the pair specific for bloodstream cells can indeed confer flagellum attachment when expressed in the procyclic cells. Hence, the flagellum attachment zone present in bloodstream cells does not have to be remodelled during differentiation. These observations were made possible due to using cell lines harboring two independent induction systems. Using flagellum attachment as a readout provided quantitative data with a straightforward interpretation of results. Text is in general well written, but the results could be discussed in a broader context, see below.

Comments for the author

Major comments

1/ The work is based on the assumption that the bloodstream FLA-FLABP are stably incorporated into a formed flagellum attachment zone, remain there after differentiation of cells to procyclics and are not replaced by their procyclic variants. While this is plausible, it could be tested experimentally by expressing tagged bloodstream proteins in bloodstream cells and in vitro differentiating these into procyclics.

2/ The work proposes a model in which FLA and FLA-BP are delivered from Golgi to the flagellar pocket. Localization of FLA2BP to the flagellum pocket region in cells not expressing FLA2 is indeed intriguing. However, the authors present images of detergent-extracted cells, in which membranes was solubilized, yet the protein signal persists. How does the signal look in living cells or fixed cells not extracted with detergent? Is it concentrated around the flagellar pocket? Could authors image these cells at a higher magnification to check whether there is a preference of the protein to localize to particular regions of the pocket, such as at the basal body region, or the section of the flagellum within the pocket? This could hint to the delivery mechanism. Furthermore, what is the indication that FLA1 or FLA2 are also delivered to the pocket?

3/ The really interesting question is why does *T. brucei* express different sets of proteins in different life cycle stages. The authors recently published a thorough review on the flagellum attachment proteins in kinetoplastids (Trends in Parasitology, 2023), in which they discuss possible reasons including predicted structure of the proteins versus the thickness of the surface coat. While I realize that this manuscript is a short report discussing some of these ideas in the context of the presented data would be beneficial.

Minor comments

Materials and Methods

Page 6, lines 11-15 - was BglIII used for cloning FLA2 into the pJ1271 vector?

Page 6, lines 49-51 - could you please clarify how were the lines with the constitutive expression of FLA2 generated?

Figures

Figure 1 - it would be useful to show also fields of view. Explain in figure legend what do NI, DOX, VAN, and DV mean; these are explained only later in legend of Fig. 2

Figure 2C and D - can you specify which conditions are shown or were quantified (induction of expression and knock down of which proteins)?

Figure 3D and E - can you specify which conditions are shown or were quantified (induction of expression and knock down of which proteins)? Also in 3D bottom left the white arrowheads appear to be shifted downwards.

Legend of Supplementary figure 1 and 3 - the expression 'cytoskeleton cells' is confusing

Supplementary figure 2D - The existence of orthologs FLA2-1 And FLA2-2 should be explained for example in the figure legend if not possible to include in the main text. Otherwise it is difficult to understand this table.

First revision

Author response to reviewers' comments

We thank the reviewers for their supportive comments and appreciate that they have highlighted the clarity of our experimental data.

Comments from the Reviewers:

Reviewer 1: This is an excellent paper, which focuses on a relatively unexplored aspect of the biology of flagellum attachment in trypanosomes: why does *T. brucei*, unusually amongst its relatives, encode two pairs of FLA-FLABP complexes during the life cycle? The authors perform a clear and comprehensive series of experiments that demonstrate rigorously that FLA2-FLA2BP, which normally operates in flagellum attachment in bloodstream form (mammalian stage) *T. brucei*, can also operate in procyclic (tsetse stage) *T. brucei*. Furthermore, they provide evidence that the two FLA-FLABP complexes interact separately: ie there is specific and non-interchangeable interactions between FLA1-FLA1BP and FLA2-FLA2BP.

SUGGESTIONS TO AUTHORS

I have no issues with any of the experiments presented, or with the clearly articulated conclusions. Instead, I congratulate the authors on a very nice study that is, in my view, suitable for publication. However, if the authors had the space and time, perhaps they could add a line or two about a question that is not addressed in this nice work: if FLA2-FLA2BP can functionally replace FLA1-FLA1BP in flagellum attachment in PCF cells, why has *T. brucei* evolved two forms of this complex?

We thank the reviewer for raising this, and we agree that a better discussion is valuable for the paper. We have included a section briefly discussing the evolution of FLA/FLABP genes in *T. brucei*. In our previous paper (PMID: 36933967), we showed that the duplication of FLA and FLABP was only seen in *T. brucei* and *T. congolense* and we postulated that this was required to enable these parasites to maintain three distinct major surface coat proteins that are tuned for different extracellular environments. Here, we have shown that FLA2/FLA2BP can operate in a cell covered by procyclins for an extended period, suggesting that these FLA-FLABP pairings do not require a specific coat environment in which to work, though it is currently unclear whether FLA1-FLA1BP could operate in a cell covered by VSGs. However, we must bear in mind these experiments were done in vitro so the consequence of expressing the FLA2-FLA2BP pairing while in the tsetse fly is unknown. Together, this points to a potential tuning phenomenon, with each pairing tuned to operate most effectively within that coat environment (FLA2-FLA2BP with VSG; FLA1-FLA1BP with procyclins) in the context of the specific extracellular environment in the tsetse fly or mammalian host.

Reviewer 2: SUMMARY OF THE ADVANCE MADE IN THIS PAPER AND ITS POTENTIAL SIGNIFICANCE TO THE FIELD

This manuscript presents a study of proteins of the flagellum attachment zone, the structure critical for attachment of the flagellum to the cell body in the parasitic flagellate *Trypanosoma brucei*. The study focuses on pairs of interacting membrane proteins and addresses whether the pair specifically expressed in the bloodstream form of the parasite can operate in the membrane environment of a tsetse fly form, which is expected to happen at early stages of fly infection. The experiments provide clear evidence that the pair specific for bloodstream cells can indeed confer flagellum attachment when expressed in the procyclic cells. Hence, the flagellum attachment zone present in bloodstream cells does not have to be remodelled during differentiation. These observations were made possible due to using cell lines harboring two independent induction systems. Using flagellum attachment as a readout provided quantitative data with a straightforward interpretation of results. Text is in general well written, but the results could be discussed in a broader context, see below.

SUGGESTIONS TO AUTHORS

Major comments

1/ The work is based on the assumption that the bloodstream FLA-FLABP are stably incorporated into a formed flagellum attachment zone, remain there after differentiation of cells to procyclics and are not replaced by their procyclic variants. While this is plausible, it could be tested experimentally by expressing tagged bloodstream proteins in bloodstream cells and in vitro differentiating these into procyclics.

Of course, we would want to look at transition forms, but this will form part of our future work, as part of a larger programme to investigate this transition process for other life cycle paralogous pairs (CAP5.5/CAP5.5V, CAP51/CAP51V, etc.). Moreover, this is not a straightforward process, as we would need to construct sophisticated cell lines in which the proteins are tagged, and the appropriate UTRs are retained as antibodies are not available.

Our results also raise additional, broader questions that would form part of this new project. For example, we have shown that together FLA1 and FLA2BP are unable to maintain flagellum attachment. This suggests that in the transition form the new flagellum and associated FAZ would require the PCF pairing of FLA1-FLA1BP and therefore a mechanism would be needed to ensure no mixing of the PCF and BSF FLA-FLABP pairings, due to the likely dominant negative effect of FLA2BP if trafficked to new flagellum. This mechanism may involve i) rapid downregulation of FLA2 and FLA2BP (Derek Nolan has shown tight regulation of FLA2BP between BSFs and PCFs - PMID: 23335957), ii) only targeting FLA1/FLA1BP to the new flagellum, or iii) a combination of i) and ii). Overall, this problem is best answered in a wider future programme that considers why this parasite has a wider set of BSF/PCF cytoskeletal/membrane pairs.

2/ The work proposes a model in which FLA and FLA-BP are delivered from Golgi to the flagellar pocket. Localization of FLA2BP to the flagellum pocket region in cells not expressing FLA2 is indeed intriguing. However, the authors present images of detergent-extracted cells, in which membranes was solubilized, yet the protein signal persists. How does the signal look in living cells or fixed cells not extracted with detergent? Is it concentrated around the flagellar pocket? Could authors image these cells at a higher magnification to check whether there is a preference of the protein to localize to particular regions of the pocket, such as at the basal body region, or the section of the flagellum within the pocket? This could hint to the delivery mechanism. Furthermore, what is the indication that FLA1 or FLA2 are also delivered to the pocket?

In this manuscript, we used detergent extracted cytoskeletons as we wanted to examine the localisation of proteins that were stably incorporated into the FAZ. We have now repeated the induction of the FLA2BP::Ty construct only and examined its localisation in whole cells after formaldehyde fixation and permeabilization. In these cells, we can see FLA2BP::Ty is found in the cytoplasm with a bright signal around the flagellar pocket. These images have been included in Supplementary Figure 3. We examined the flagellar pocket signal in detail but were unable to discern any underlying pattern that might indicate delivery of material to a specific point on the flagellar pocket. We believe that the flagellar pocket signal observed in the detergent extracted cytoskeletons is due to an accumulation of FLA2BP::Ty around this region of the cell, which is not fully solubilised by detergent treatment.

We have been unable to successfully tag FLA1 or FLA2 with an epitope tag or fluorescent protein nor do we have any antibodies to these proteins, so we are unable to track their movements

through the cell. However, as the flagellar pocket is the only site for exocytic vesicles to fuse to the cell membrane and as these proteins have a signal peptide and transmembrane domain, it is highly likely that FLA1 and FLA2 will be delivered to the flagellar pocket before being assembled into the FAZ.

3/ The really interesting question is why does *T. brucei* express different sets of proteins in different life cycle stages. The authors recently published a thorough review on the flagellum attachment proteins in kinetoplastids (Trends in Parasitology, 2023), in which they discuss possible reasons including predicted structure of the proteins versus the thickness of the surface coat. While I realize that this manuscript is a short report discussing some of these ideas in the context of the presented data would be beneficial.

Reviewer 1 raised a similar point, and we have now included a discussion about the duplication and diversification of these proteins in *T. brucei* and how we believe this points to a tuning phenomenon, with each pairing tuned to operate most effectively within that specific coat environment and its extracellular context.

Minor comments

Materials and Methods

Page 6, lines 11-15 - was BglIII used for cloning FLA2 into the pJ1271 vector?

Yes, it was digested with *HindIII* and *BglIII* to allow cloning of FLA2 into pJ1271 while excluding the Ty sequence. We thank the reviewer for pointing that out, and we revised the text for clarity.

Page 6, lines 49-51 - could you please clarify how were the lines with the constitutive expression of FLA2 generated?

After reviewing this comment, we realised that we had in fact replaced the FLA2 3' UTR with a truncated aldolase 3' UTR that supports expression for both BSFs and PCFs. We have corrected this in the manuscript. To do this, we amplified the intergenic sequence between the gene encoding the fluorescent protein, the G418 coding sequence and its 3' UTR from the pPOT-G418 plasmid using primers that contain sequences homologous to the 3' end of the FLA2 gene and the 5' end of the FLA2 3' UTR. This construct was then transfected into the cells to enable the displacement of the FLA2 3' UTR. After selection with G418, the expression of FLA2 was confirmed by transcriptomics and the cell lines were used for the double-induction experiments. We revised this section on the manuscript to make this step clearer.

Figures

Figure 1 - it would be useful to show also fields of view. Explain in figure legend what do NI, DOX, VAN, and DV mean; these are explained only later in legend of Fig. 2

The fields of view have been included in Supplementary Figure 1A. We have now explained in the figure legend of Figure 1 the meaning of NI, DOX, VAN and DV.

Figure 2C and D - can you specify which conditions are shown or were quantified (induction of expression and knock down of which proteins)?

We have now included which proteins were depleted or expressed.

Figure 3D and E - can you specify which conditions are shown or were quantified (induction of expression and knock down of which proteins)? Also in 3D bottom left the white arrowheads appear to be shifted downwards.

We included which proteins were depleted or expressed, and the arrowheads in figure 3D were corrected.

Legend of Supplementary figure 1 and 3 - the expression 'cytoskeleton cells' is confusing

We replaced "cytoskeleton cells" with "detergent-extracted cytoskeletons" in the legends for both Supplementary Figures 1 and 3.

Supplementary figure 2D - The existence of orthologs FLA2-1 And FLA2-2 should be explained for example in the figure legend if not possible to include in the main text. Otherwise it is difficult to understand this table.

We clarified in the Figure legend of S2 that FLA2, FLA1BP and FLA2BP have two copies each in the *T. brucei* genome.

Second decision letter

MS ID#: jcs.264370R1

MS Title: Trypanosome bloodstream-specific flagellum attachment proteins can operate in an insect surface coat environment

Authors: Laryssa Vanessa de Liz; Hannah Pyle; Patr cia Hermes Stoco; Jack D Sunter

Article Type: Short Report

Dear Dr Sunter,

I am happy to tell you that your manuscript has been accepted for publication in Journal of Cell Science, pending standard publication integrity checks.